



# Impact of numerical solution approach of a plant hydrodynamic model on vegetation dynamics

Yilin Fang[1], Ruby Leung[1], Ryan Knox[2], Charlie Koven[2], and Ben Bond-Lamberty[1]

[1]Pacific Northwest National Laboratory, Richland, Washington, USA

[2]Lawrence Berkeley National Laboratory, Berkeley, California, USA

*Correspondence to*: Yilin Fang (yilin.fang@pnnl.gov)

**Abstract.** Numerous plant hydrodynamic models have started to be implemented in vegetation dynamics models, reflecting the central role of plant hydraulic traits in driving water, energy and

carbon cycle, as well as plant adaptation to climate change. Different numerical approximations of the governing equations of the hydrodynamic models have been documented, but the numerical accuracy of these models and its subsequent effects on the simulated vegetation function and dynamics have rarely been evaluated. Using different numerical solution methods (including implicit and explicit approaches) and vertical discrete grid resolutions, we evaluated

the numerical performance of a plant hydrodynamic module in the Functionally Assembled Terrestrial Ecosystem Simulator (FATES-HYDRO) based on single point and global simulations. Our simulation results showed that when near-surface vertical grid spacing is coarsened (grid size > 10 cm), the model significantly overestimates above ground biomass (AGB) in most of the temperate forest locations, and underestimates AGB in the boreal forest

locations, as compared to a simulation with finer vertical grid spacing. Grid coarsening has a small effect on AGB in the tropical zones of Asia and South America. In particular, coarse surface grid resolution should not be used when there are large and prolonged water content difference among soil layers at depths due to long dry season duration and/or well-drained soil, or when soil evaporation is a dominant fraction of evapotranspiration. Similarly, coarse surface

grid resolution should not be used when there is lithologic discontinuity along the soil depth. This information is useful for uncertainty quantification, sensitivity analysis, or training surrogate models to design the simulations when computational cost limits the use of ensemble simulations.



## 1 Introduction


Vegetation plays a central role in water, energy and carbon cycle [*Arora*, 2002; *Gerten et al.*, 2004; *Levis et al.*, 2000] through the bidirectional interactions between climate and terrestrial biota. Stomatal conductance is one of plants' physiological properties that form the basis of evapotranspiration parameterizations in physically based hydrological models [*Arora*, 2002] and

Earth system models (ESMs). Soil moisture plays a vital role in regulating stomatal conductance and plant water status [*Anav et al.*, 2018; *Buckley*, 2019]. How ESMs represent soil moisture regulation on stomatal conductance thus has important implications for the partitioning of evapotranspiration into evaporation and transpiration, the soil moisture profiles that influence soil hydrological processes, and plant growth and vegetation dynamics as well as the accurate

simulation of land-atmosphere energy and water fluxes.

Most ESMs use non-mechanistic soil moisture stress parameterizations that relate a metric of soil moisture status to attenuation of stomatal conductance in response to declining soil water under drying conditions, ignoring vegetation water use strategies [*Kennedy et al.*, 2019]. The ESM community has worked to replace such empirical water stress parameterizations with more

realistic mechanistic plant hydrodynamic representations. Water transport in the soil-plant-atmosphere continuum is often represented using a Richard's type equation in the mixed-form or potential-based form, which has been commonly used to describe fluid flow in partially saturated porous media [*Celia et al.*, 1990; *Lehmann and Ackerer*, 1998]. In the mixed-form the equation is written using both water potential and water content as the dependent variables, while the

equation is written using water potential as the dependent variable in potential-based form. Hydrodynamic representations are nonlinear problems, because xylem hydraulic conductivity ($K_s$) and plant water storage vary nonlinearly with water potential in each organ in the model, so they are typically solved numerically.

Different numerical approaches, with various degrees of simplifications, have been used in

the literature to solve the equations in the plant hydrodynamic models. Hydraulic models that consider water storage in the simulated plant organs may use numerical techniques that feature non-iterative (e.g., explicit time integration) or iterative approaches (e.g., Newton's method for nonlinear problems). Examples of models using non-iterative solution approach are the Soil Plant Atmosphere (SPA) model [*Williams et al.*, 1996], a dynamic water flow and storage





model called HydGro [*Steppe et al.*, 2006], the trait forest simulator (TFS) [*Christoffersen et al.*, 2016], ED2-hydro [*Xu et al.*, 2016], and Noah-MP-PHS [*Li et al.*, 2021]. Models that use iterative solutions include FETCH2 [*Mirfenderesgi et al.*, 2016], the soil plant continuum model [*Sperry et al.*, 1998; *Sperry et al.*, 2016], and a porous media model for the hydraulic system [*Chuang et al.*, 2006]. There has however been no systematic evaluation and comparison of their

model performance and their consequential impact on evapotranspiration partitioning, soil moisture dynamics, and vegetation function and dynamics simulated by the ESMs.

As key differences among different plant hydrodynamic models lie in the numerical approaches used to solve the plant hydrodynamic equations, we implement several numerical solution options for the hydrodynamic problems in the same model to facilitate comparison. The

model used here is the plant hydrodynamic model in the Functionally Assembled Terrestrial Ecosystem Simulator (FATES-HYDRO) for illustrations. We compare the model performance of the various options and their impacts on simulating evapotranspiration partitioning, soil moisture dynamics, and vegetation dynamics. Our focus is on two aspects of the numerical solutions: vertical grid aggregation of the soil column and use of explicit vs. implicit solvers of the

hydrodynamics equations, as they have implications for the accuracy and computational efficiency of the numerical solvers.

## 2   Model description

### 2.1 Functionally Assembled Terrestrial Ecosystem Simulator (FATES)

FATES is a vegetation demographic model, which uses the Ecosystem Demography (ED)

[*Moorcroft et al., 2001*] and Perfect Plasticity Approximations (PPA) [*Purves et al.*, 2008] to scale from cohorts of individual plants of different plant functional types growing within a mosaic of patches with different disturbance histories to the land surface [*Fisher et al.*, 2018; *Koven et al.*, 2020]. FATES has been coupled to the Energy Exascale Earth System Model (E3SM) Land Model (ELM) [*Caldwell et al.*, 2019; *Leung et al.*, 2020], which we use here.

Processes that are simulated in FATES include physiological processes on 30 min time steps, which include photosynthesis, respiration, and radiative transfer, as well as land-surface energy balance and all plant-soil hydrologic calculations coordinated with the land-surface model. At daily timescale, FATES handles plant growth, mortality, and disturbances. More details of


FATES can be found in Fisher et al. [2015] and Koven et al. [2020], as well as in the online

documentation https://fates-docs.readthedocs.io/en/latest/fates_tech_note.html.

The Energy Exascale Earth System Model (E3SM) is an Earth system model containing components for atmosphere, land, ocean, sea ice, and river [*Caldwell et al.*, 2019; *Leung et al.*, 2020]. The land model in E3SM, referred to as ELM, was based on the Community Land Model version 4.5 (CLM4.5) [*Oleson et al.*, 2013]. The E3SM land model for this study is similar to the

Community Land Model version 4.5 [*Oleson et al*., 2013] except for some biogeochemistry components [*Ricciuto et al*. 2018; *Burrows et al.*, 2020] and a one-dimensional variably saturated subsurface flow model [*Bisht et al.*, 2018], which were not turned on in this study. In ELM, the soil hydraulic properties are assumed to be a function of sand and clay contents based on the work by Clapp and Hornberger [1978] and Cosby et al. [1984], and soil organic properties

[Lawrence and Slater 2008]. The bulk hydraulic properties are weighted averages of the properties of the soil mineral and organic contents, and details can be found in Oleson et al. [2013]. As described in Oleson et al. [2013], the mineral soil texture dataset for each soil layer was created from the International Geosphere-Biosphere Programme (IGBP) soil dataset (Global Soil Data Task 2000) of 4931 soil mapping units and their sand and clay content [*Bonan* et al.

2002]. The majority of the globe soil organic matter data is from ISRICWISE [*Batjes*, 2006], and those from the high latitudes come from the 0.25º version of the Northern Circumpolar Soil Carbon Database [*Hugelius* et al. 2012]. Both datasets report carbon down to 1m depth and carbon is partitioned across the top seven soil layers as in Lawrence and Slater [2008].

**2.2 FATES-HYDRO**

FATES-HYDRO is an extension of the plant hydrodynamic model described in Christoffersen et al. [2016]. It solves transient water flow from soil to roots, stem and leaf to meet the transpiration demand. Xylem transport in FATES-HYDRO follows Darcy's law, which says that flow rate in the porous media is proportional to the hydraulic gradient and the hydraulic conductivity. FATES-HYDRO accounts for the plant internal water storage that can buffer the

imbalance of root water uptake and transpiration demand. In discretized approximation, the transient water mass balance equation along the hydraulic path for each node *i* can be written as:

$$\rho_w V_i \frac{d\theta_i}{dt} = \sum_{j=1}^{k} Q_{i,j} \qquad (1)$$



where $i$ is the node number and $i$ at the leaf node is equal to 1, with nodes ordered from top to bottom and horizontally from the root node to soil node (Fig.1 ). Discrete fluxes between the

compartment of interest and a total of k other connected compartments are indexed by j. k is 1 for the leaf node, and it is equal to 2 for compartment other than the transporting root compartment where $k$ equals the number of soil layers plus 1. $\rho_w$ is the density of water (kg m⁻³), $V_i$ is the volume of modeled compartment or node (m³), $t$ is time (s), $\theta_i$ is water content (dimensionless), $Q_{i,j}$ (kg s⁻¹) is the water mass flux between compartments $i$ and $j$ (positive for

movement towards the leaf).

$$Q_{i,j} = -K_i\big(\rho_w g(z_i - z_j) + (\psi_i - \psi_j)\big) \qquad (2)$$

The flux over a connection is driven by potential differences between compartments, where $g$ is acceleration due to gravity (9.81 m s⁻²) and $\psi_i$ is xylem or soil matric water potential (MPa),

which is calculated based on pressure-volume curve, analogous to the soil water retention curve in ELM soil hydrology [*Christoffersen et al.*, 2016]; $z_i$ is the elevation above (positive) or below (negative) the ground (m), and $K_i$ is the conductance (kg Mpa⁻¹ s⁻¹) at the boundary between compartments $i$ and $j$. $K_i$ is calculated as the product of the relative hydraulic conductance $k_{r,i}$ (dimensionless) and the maximum conductance (kg mPa⁻¹ s⁻¹) at the boundary of nodes $i$. Note

the maximum conductance is a product of the conduit cross-section and the material conductivity. Relative conductance, $k_{r,i}$, is calculated by the vulnerability curve using an inverse polynomial function [*Manzoni et al.*, 2013] in plant compartment as follows:

$$k_{r,i} = \left[1 + \left(\frac{\psi_i}{P_{50,i}}\right)^{a_i}\right]^{-1} \qquad (3)$$

$P_{50}$ is the water potential leading to 50% loss of hydraulic conductivity, $a_i$ is a shape index

(dimensionless). The water stress function is usually empirically represented in land models as a function of soil water matric potential, but here is replaced by an empirical function of leaf water potential to include the hydraulic impacts on stomatal conductance [*Christofferson et al.* 2016]:

$$\beta = \left[1 + \left(\frac{\psi_l}{P_{50,gs}}\right)^{a_{gs}}\right]^{-1} \qquad (4)$$





where $\beta$ is a water stress fraction, , $\psi_l$ is the leaf water potential (MPa), $P_{50,gs}$ is the leaf water

potential $\psi_l$ (MPa) at 50% stomatal closure, and $a_{gs}$ is the shape parameter (dimensionless).

FATES-HYDRO divides each individual tree into four compartments: leaf, stem, transporting root (troot), and absorbing root (aroot) as shown in Figure 1.  In this study, all compartments except for the absorbing root are represented by a single node for each in the discrete approximation of the equation. The absorbing root is discretized into the same number

of nodes as the number of soil layers for soil hydrology in ELM.  The soil in each layer is radially discretized into cylindrical shells representing the rhizosphere around an absorbing root (Fig. 1).

## 2.3 Numerical solutions

We provide the following options to solve Equation 1, including non-iterative and iterative

approaches. For the non-iterative approach, as the time step in FATES for fast processes is 30 min, we use a sub-stepping time integration, with a sub-time step of 10 min, following the timestep used in ED2 [*Xu et al.*, 2016]. Nonlinear iterative methods, including the Newton and Picard schemes, are commonly used to solve Richards' equation [*Albuja and Avila*, 2021; *Brenner and Cances*, 2017; *Caviedes-Voullieme et al.*, 2013; *Celia et al.*, 1990; *Lehmann and*

*Ackerer*, 1998; *List and Radu*, 2016]. The Picard scheme is a globally convergent method with a low solution efficiency because of its first-order convergence rate. On the other hand, the Newton method is only locally convergent, but a converged solution is not always guaranteed. In this study, we use the Newton method.

We use water content $\theta$ in each compartment as unknowns for the Newton iteration. Coupled

with a backward Euler approximation in time, the residual form of Eq. 1 for each compartment is defined as

$$Re_i = \rho_w V_i \frac{\theta_i^{n+1,m+1} - \theta_i^n}{\Delta t} - \sum_{j=1}^{k} Q_{i,j}^{n+1,m+1} \qquad (5)$$

Superscripts $n$ and $m$ denote time level and iteration number. The correction quantity $\delta$ of $\theta$ at each point from the last iteration is


$$\delta^m = \theta^{n+1,m+1} - \theta^{n+1,m} \qquad (6)$$

$\delta$ is the solution of the following matrix equation

$$[A]\{\delta\} = -[Re] \tag{7}$$

where $A$ is the Jacobian matrix calculated from the derivative of the non-linear function in Eq. 5 with respect to the unknown water content at each point, and each row in Eq. 7 is $(A\delta)_i = \sum_{j=1}^{k} C_j \delta_j$; $C_j = \frac{\partial Re_i}{\partial \theta_j^{n+1,m}}$. Taking compartment $i$ connected to compartments $i\text{-}1$ and $i\text{+}1$ as an example, and expanding the water flux $Q^{n+1,m+1}$ in a truncated Taylor series with respect to water content $\theta$ at the expansion point $\theta^{n+1}, . i. e.,$ we obtain

$$Q^{n+1,m+1} = Q^{n+1,m} + \frac{dQ}{d\theta}\big|^{n+1,m}(\theta^{n+1,m+1} - \theta^{n+1,m}) + 0(\delta^2) \tag{8}$$

Neglecting the higher order terms, the i$^{\text{th}}$ row in Eq. 7 becomes

$$\frac{\partial Q_{i-1}^{n+1,m}}{\partial \theta_{i-1}^{n+1,m}} \delta_{i-1}^m + \frac{\rho_w V_i}{\Delta t} \delta_i^m + \frac{\partial Q_{i-1}^{n+1,m}}{\partial \theta_i^{n+1,m}} \delta_i^m - \frac{\partial Q_i^{n+1,m}}{\partial \theta_i^{n+1,m}} \delta_i^m - \frac{\partial Q_i^{n+1,m}}{\partial \theta_{i+1}^{n+1,m}} \delta_{i+1}^m = Q_i^{n+1,m} - Q_{i-1}^{n+1,m} -$$

$$\rho_w V_i \frac{\theta_i^{n+1,m} - \theta_i^n}{\Delta t} \tag{9}$$

Equation 7 is solved during each iteration. Convergence of the Newton iteration is achieved when the maximum residual is less than $10^{-8}$ or when the following inequality is satisfied at all nodes i:

$$\delta_i^m < \tau \tag{10}$$

where $\tau$ is the specified tolerance/accuracy. If the scheme is not convergent within the specified maximum number of iterations during a time step, Eq. 1 is explicitly integrated using sub-time stepping within each time step such that the Courant-Friedrichs-Lewy condition [*Courant et al.*, 1928] is below 1.0.

The stack of vertical soil-root interaction layers can be customized by the user to save computation time or carry out a grid convergence study, where a series of grids are generated and model computations are performed to analyze the differences among the results with each grid configuration. In our model configuration, the top soil layer thickness can be as thin as a few centimeters.

Boundary conditions for the system include transpiration flux through leaves and zero-flux for the outermost rhizosphere element assuming the rhizosphere shells encompass the whole soil





layer. The rate of water mass change in each soil layer during a time step of FATES-HYDRO is
passed to the land model as a source/sink term to calculate the soil water state for the next time
step. This rate differs from the transpiration sink as water can be stored or lost in the
compartments.

**2.4 Grid aggregation**

In the default model setting, there are a total of 10 soil layers. Soil layers are the discrete
vertical interval over which ELM resolves water content. ELM updates water content via
processes of vertical percolation, infiltration, evaporation, and through runoff and drainage of
uppermost and lowermost layers respectively. The water content in each of these layers is
presented as an initial condition to FATES-HYDRO. The grid thickness varies from 1.7 cm at
the top layer to 1.5 m at the bottom layer. The thickness for layers 2, 3, 4, 5 is 2.76 cm, 4.55 cm,
7.5 cm, and 12.3 cm, respectively. To reduce computation time and avoid potential numerical
stability issues caused by the thin layers, the FATES-HYDRO model can be configured such that
several soil layers are aggregated to solve for a fewer number of equations. We define a
"rhizosphere layer" as a discrete vertical interval that may contain one or more discrete soil
layers, over which the water contents and the fluxes in fine-root tissues are resolved. For
simplicity, the depth of the first rhizosphere layer for FATES-HYDRO aligns with the depth of
the last soil layer that's been aggregated, and the rest of the rhizosphere layer thickness is the
same as those from ELM at the same depth. For example, as shown in Figure 2, if the first 4 soil
layers (s1 to s4) in ELM are aggregated to form the first rhizosphere layer r1 in FATES-
HYDRO, the thickness of r1 is the sum of the thickness of s1 to s4, and the thickness of r2 is the
same as s5, and so on. Total water mass in s1 to s4 are assigned to r1. After FATES-HYDRO is
solved, the flux exchange between the root and the rhizosphere for r1 is proportionally assigned
to s1, s2, s3, and s4 weighted by the product of soil layer thickness and hydraulic conductivity of
s1 to s4.


**3    Simulation Experiments**





Global and point-scale simulations were performed to assess the impact of vertical soil layer aggregation. A 4×5 degree resolution global simulation was run for 100 years with two rhizosphere grid configurations: 1) no soil layer aggregation, i.e., rhizosphere soil layers in FATES-HYDRO are the same as ELM soil layers, and 2) aggregating the top 5 ELM soil layers. A repeating cycle of a three-year (2000-2002) atmospheric forcing data from Qian et al. [2006] is used to drive the model.

Four locations were selected after analyzing the global simulation to further evaluate model performances using different approaches. For point-scale at selected locations, simulations with aggregation of 1, 3, 5 and 7 layers were first run using the implicit approach to check for model differences in AGB. If large differences were found between simulations, extra simulations of different layer aggregations for some points were run to determine which scheme starts to cause large difference and the relative computation costs. Each point was also simulated using the explicit approach for comparison with the implicit approach.

### 3.1 Global simulation

It takes longer time to solve more equations. The wall clock time for the simulation using no aggregation is 1.5 times of that for the simulation using 5-layer aggregation. The difference in above ground biomass (AGB) using different layer aggregation strategies varies by regions, regardless of the total number of simulation years (Fig. 3). It took about 20 days using 120 processor cores to complete 100-year simulation for the simulation without layer aggregation. Model differences with and without soil layer aggregations were evident during a much earlier simulation year, for example year 15.

We found that when more rhizosphere soil layers near the surface are aggregated, the model significantly overestimates AGB (negative ΔAGB in Fig 3b) in most of the temperate forest locations and underestimates AGB in the boreal forest locations relative to the simulation in which soil layers are not aggregated. Layer aggregation has only small effects on AGB (< 5%) in tropical zones near Asia and South America. ΔAGB follows the same pattern as the differences in ET (ΔET) (Fig. 3c). In general, regions with large ΔAGB have small AGB. In the southern hemisphere where ΔAGB is high, the annual mean of soil water saturation in the soil layer at the



ground surface is generally lower than that in the soil layer 17 cm (layer 5) below the surface (negative soil water saturation differences between soil layer 1 and layer 5 ($\Delta Sl_{15}$) in Fig. 3d) and the opposite (positive $\Delta Sl_{15}$) is true in a large fraction of the northern hemisphere. That is, mixing of soil water from layers of contrasting water saturation when aggregating grids is the

main cause of $\Delta$AGB.

Negative soil water saturation differences $\Delta Sl_{15}$ between the shallow and deep soil layers can be caused by long dry season durations and/or when the soil is well-drained (rapid decrease of water content with matric potential in the capillary region); regions with large $\Delta$AGB exhibit low clay content and/or long duration of dry seasons (Fig. 4). The dry season duration is calculated as

the number of months when evapotranspiration is larger than precipitation. For example, $\Delta$AGB is big in the temperate forest regions which exhibit large organic matter density compared to the deeper soil layers (Fig. 4f), but the soils in those regions mostly have low and relatively homogeneous clay content (Fig 4c,e). $\Delta$AGB in Amazon is small because of the high clay content (> 30%) and short dry season durations.

In the high latitudes, layer aggregation schemes can still cause large difference in AGB even in places with high clay content and short dry season duration because frozen soil can cause large water content differences in surface soil layers. Ice in the soil can greatly decrease the hydraulic conductivity of the soil through a power law form of the ice-impedance factor, leading to nearly impermeable soil layers [*Swenson et al.*, 2012]. A large fraction of the high latitudes has high

ratios of soil evaporation to evapotranspiration ratio (E/ET) (Fig. 4b). E is determined by the near surface soil water states, and a large ratio of E/ET can cause significant water content difference in soil layers. Therefore, the simulated AGB will be significantly changed if the surface soil is aggregated with the deeper wetter soil. Note that this simulation is not calibrated, thus the high E/ET ratio at the high latitudes may be overestimated.


## 3.2 Interpretation of the model difference by machine learning

To confirm the factors such as E/ET ratio and soil property discontinuity along depth are the driving factors for the model differences when aggregating grids in the global simulations, we calculated $\Delta$AGB between the results from the simulation using no layer aggregation and the 5-

layer aggregation, averaged from the last five years of the simulation, and classified the grids with difference greater than 5% as "Positive Difference" (i.e., more AGB from no grid





aggregation), less than 5% as "Negative Difference", and the rest as "Comparable". We then constructed a machine learning model to evaluate the classification skills using the XGBoost classifier from the scikit-learn package in Python and model explanation using SHapley Additive

exPlanations (SHAP) by providing impact of features on individual predictions [*Lundberg and Lee*, 2017]. We developed a model using the following inputs including environmental variables: surface elevation, clay content in soil layers 1 to 5 (clay_l1, clay_l2, clay_l3, clay_l4, and clay_l5), clay content difference between the top 1 and the average of the top 5 layers (dc1c5), organic matter (OM) density in soil layers 1 to 5 (org_l1, org_l2, org_l3, org_l4, and org_l5) and

the OM density difference between the top 1 and the average of the top 5 layers (do1o5), precipitation, and temperature, and model dependent variables: soil evaporation-to-evapotranspiration ratio (efrac), dry season duration (mon_dry), soil water saturation from the top five soil layers near the ground surface (sw1, sw2, sw3, sw4, sw5). Clay content and organic matter density were selected as features because they determine hydraulic conductivity. Model

dependent variables were selected to understand the physical process drivers of modeled AGB discrepancy. The machine learning classifier accuracy for the training and test data set are 87% and 67%, respectively (Figure 5). Though not so good according to a general rule of thumb, both training and test data exhibit consistent feature importance.

       SHAP feature importance confirmed some of our previous hypothesis explaining the model

differences. Dry season duration (feature mon_dry) has relatively small importance in explaining the negative model difference. The top 5 SHAP values for negative model differences in AGB are dc1c5, do1o5, precp, sw5, and sw2, while those responsible for positive model differences are dc1c5, sw4, precp, temp, and org_l1. Temperature becomes important because it affects the presence of soil ice in high latitudes, which affects soil hydraulic conductivity. Features efrac,

dc1c5, precp, org_l1, and elev are important explaining small model differences in AGB. Because of the dependencies of efrac and mon_dry on soil moisture and soil hydraulic conductivity (affected by soil texture and ice), it is not surprising that soil water saturation in deep soil layer is important explaining the model differences. The deep soil water status can affect soil wetness in the rhizosphere soil shell when there is large contrast between the soil

water content simulated by ELM between the top and deep soil layers.

### 3.3 Single point simulations



To further understand the effect of soil layer aggregation, we selected a point in the tropical zone (P1, (10º N, 80º W)), temperate zone (P2, (46º N, 95º W)), polar zone (P3, (66º N, 15º E)),

and equatorial zone (P4, (6º S, 135º E)), respectively from the global simulation and ran a one-hundred year simulation subjecting to a repeating cycle of a three-year (2000-2002) atmospheric forcing from Qian et al. [2006] at each selected location (Fig. S1). Default FATES-HYDRO parameters are used without modification. Different rhizosphere grid configurations and numerical schemes were run and compared for each point. The clay content and organic matter

density at each point are listed in Table S1. At P1 to P3 the clay content is around 30%, 36%, and 21%, respectively, and it varies from 35% to 26% from the top to the bottom of soil at P4. Organic matter density varies the most with depth at P3.

### 3.3.1    Aggregation schemes

At the end of the simulation, the fraction of wall clock time of simulations at each point using 3, 5, and 7 layer aggregations are around 0.8, 0.7, and 0.5 times of the that from the simulation with no layer aggregation.

AGB at point P1 starts to show significant difference (49.3% on average compared to no aggregation) when only two rhizosphere layers are simulated, i.e., aggregating the top 9 layers

for the surface soil (Fig. 6). For P2, aggregating 5 layers and more can result in more than 12% of AGB difference compared to no aggregation. The same is true for points P3 and P4, with larger differences for more layer aggregation. This kind of AGB difference between different layer aggregation schemes show up early in the simulation as shown in Figure 7 for the 10-year simulation comparison. This means one does not need to run the full simulation to test whether

layer aggregation will cause large AGB errors if computation cost is a concern.

At P1, the largest difference in water content is in February, the driest month, while the difference is trivial in the other months (Fig.8). Because the dry season duration is short, and clay content is relatively homogeneous at P1, aggregating the surface layers at this point does not

cause large difference in AGB. Layers 4 and deeper at P2 and P3 are affected by ice impedance, creating large difference from the top 3 layers. The water content at P3 is also affected by the large contrast in organic matter density between the surface layer and deeper soil from layer 4.



At P4, lithologic discontinuity (clay content separation) between the top 3 layers and bottom
layers can cause inaccuracy in soil water content, hence AGB.


### 3.3.2 Integration Methods

Implicit and explicit integrations of Eq. 1 for points P1 to P4 were run to evaluate model
performance and computation costs. The simulations were performed without layer aggregation
for comparison of the integration schemes. The time step for the explicit integration is 10 min.

There are discrepancies between the two integration approaches at P1, but results show less than
2% AGB difference at the end of the simulation year (Fig. 9). Results at P2 to P4 are almost
identical. However, simulations took more time using the explicit integration approach, with wall
clock times 1.85, 1.31, 1.93, and 1.72 times of that of the implicit integration for P1 to P4,
respectively.

The explicit approach is easier to implement than the implicit approach in terms of coding.
However, the explicit approach tends to have stability issue and requires small time steps, while
the implicit approach is stable using large time steps but may require many iterations to converge
to a solution. These numerical experiments with different integration schemes here can serve as
benchmark against each other. In the meantime, it shows that the 10-min time step in ED2 [*Xu et*
*al.*, 2016] is a reasonable time step for these tests, but it is always a good practice to do
convergence and stability tests for a specific study.

## 4 Conclusions

We have implemented multiple numerical schemes in solving plant hydrodynamic equations,
including explicit and implicit iterative integration of Eq. 1, as well as aggregating rhizosphere
soil layers for the considerations of computation cost and numerical difficulties. While not
exhaustive, our results showed that explicit integration using a 10-min time step results in
comparable AGB with the implicit method, but takes longer simulation time. We also found that
care should be taken when configuring soil layering as it can significantly affect AGB results.
Large water content differences among soil layers at depth can occur due to lithologic
discontinuity, long dry season duration, high E/ET ratio, or well-drained soil. Short time
simulation tests can be sufficient to evaluate how model configurations or numerical approaches
will affect the simulated AGB accuracy. The cost and accuracy using alternative grid aggregation



methods (e.g., fewer number of cylindrical shells), and the approach to pass flux from aggregated layers back to ELM soil layers can be further investigated in the future. The results from our

analysis are useful for uncertainty quantification, sensitivity analysis, or training surrogate models to design the simulations when computation cost is limiting the selection of ensemble simulations.

*Code Availability*. The FATES-HYDRO code is available at

https://doi.org/10.5281/zenodo.6461878.

*Author contributions.* YF, RK, CK developed the code. YF set up the model, performed simulations and prepared the figures. YF, LRL, RK, CK, BB contributed to writing and editing.

*Competing interests.* The authors declare that they have no conflict of interest.

### Acknowledgements

This research was supported by the U.S. Department of Energy Office of Science Biological and Environmental Research through the Earth System Development program as part of the Energy Exascale Earth System Model (E3SM) project. The Pacific Northwest National Laboratory (PNNL) is operated for DOE by Battelle Memorial Institute under contract DE-AC05-76RLO1830.

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



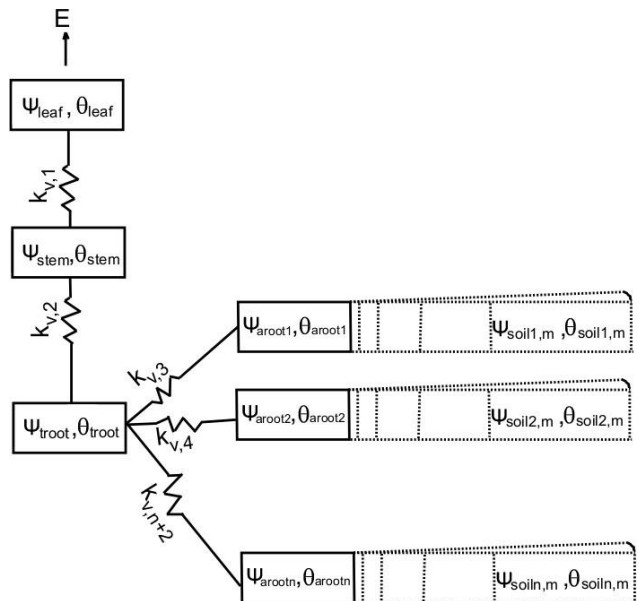

**Figure 1.** Schematic of FATES-hydro, with each box representing a compartment of plant tissue or soil rhizosphere.





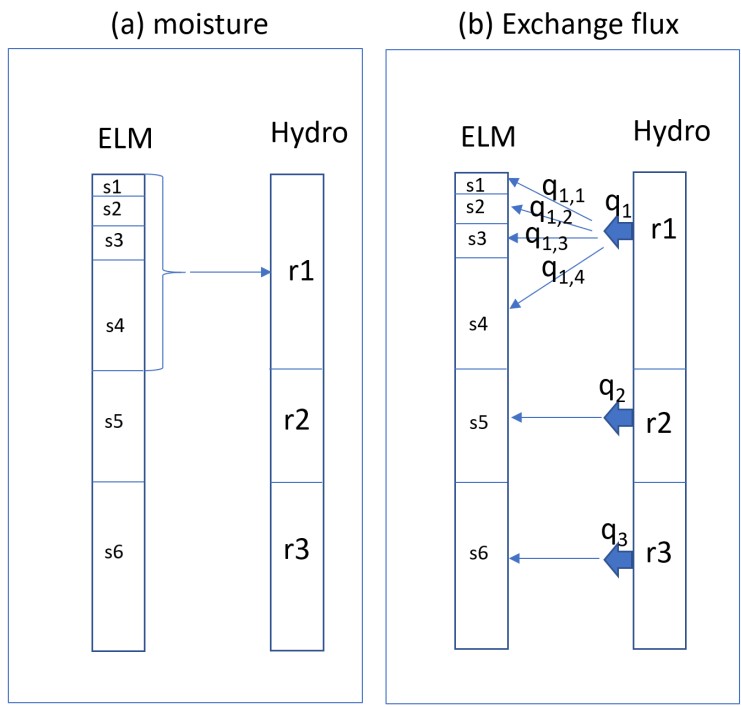

**Figure 2**. Mapping of soil water mass (a) and flux exchange (b) between the soil column in ELM and the rhizosphere in FATES-HYDRO. s stands for ELM soil layer, r stands for rhizosphere layer, q is flux exchange.

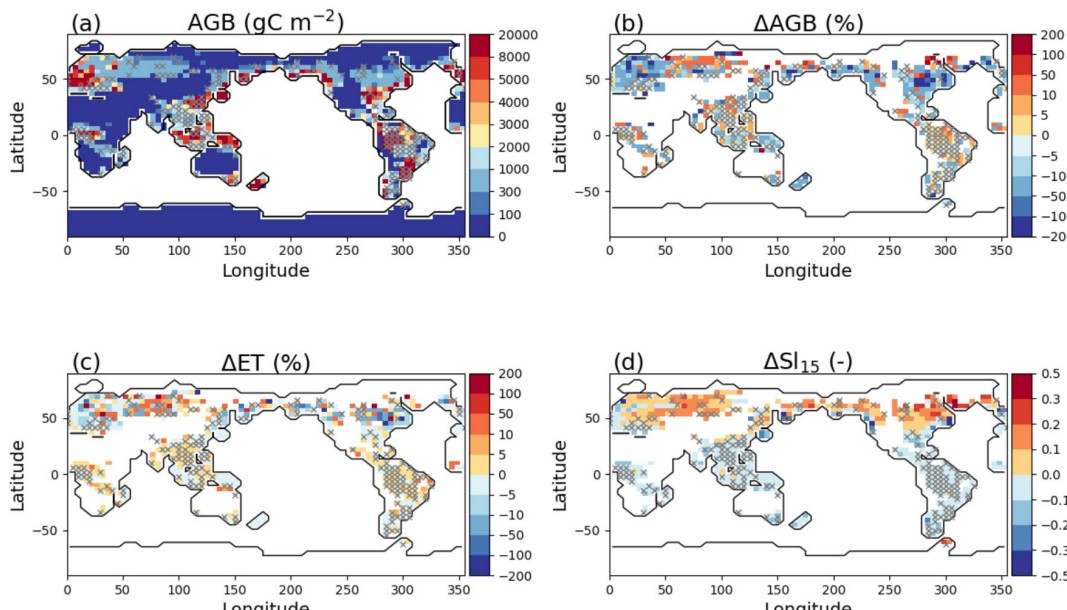

**Figure 3**. Average AGB in simulation year 100 (a), model difference resulted from layer aggregations: percent change of AGB (b) and ET (c), and average soil water saturation between soil layer 1 and layer 5 in simulation year 100 (d) without layer aggregation, The pixels in white on land in (b,c,d) have values beyond the limits of the legends, associated with AGB < 0.5 gC m$^{-2}$. Pixels with symbol × have ΔAGB less than 5%.





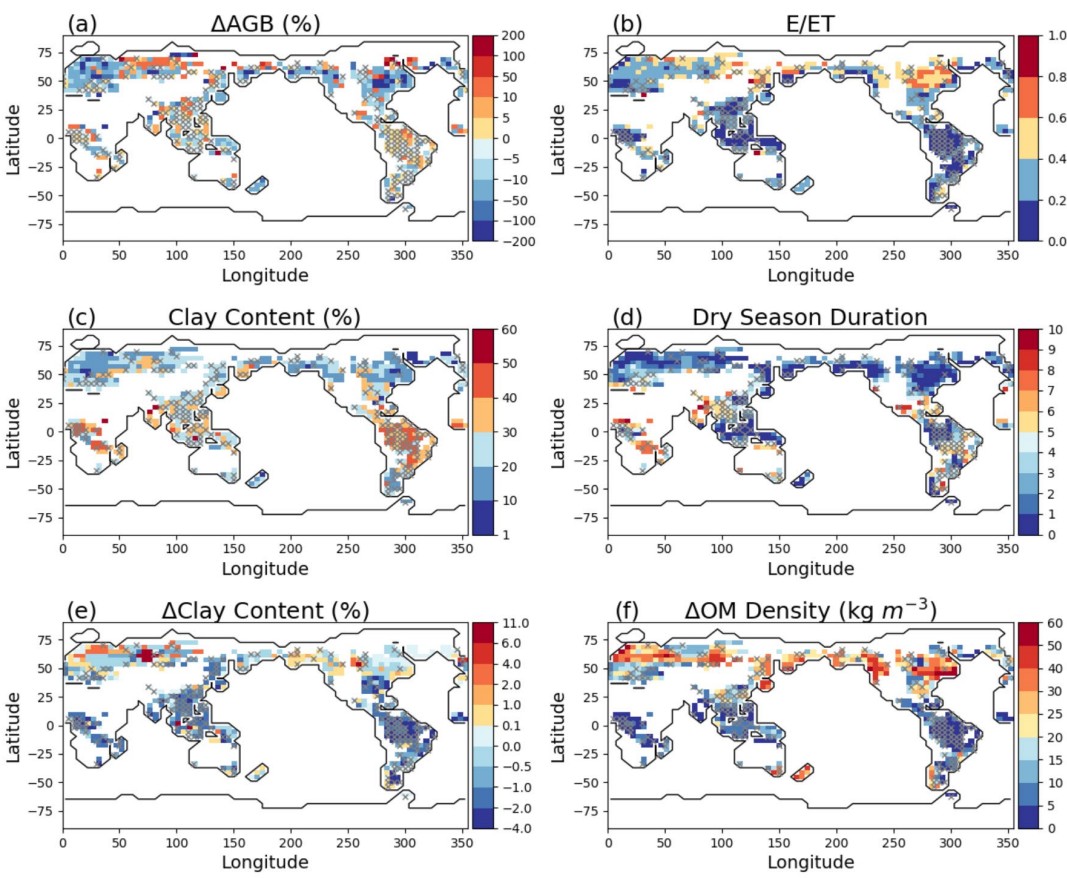


**Figure 4**. Model differences resulted from layer aggregations: percent change of AGB between no layer aggregation and aggregating 5 layers (a), and E/ET (b) for simulation year 100, average clay content in the soil column (c), and dry season durations (months) (d), clay content difference (e) and organic matter difference (f) between layer 1 and the average of the top 5
layers from the surface. The pixels in white on land have values beyond the limits of the legends, associated with AGB < 0.5 gC m⁻². Pixels with symbol × have AGB differences less than 5%.





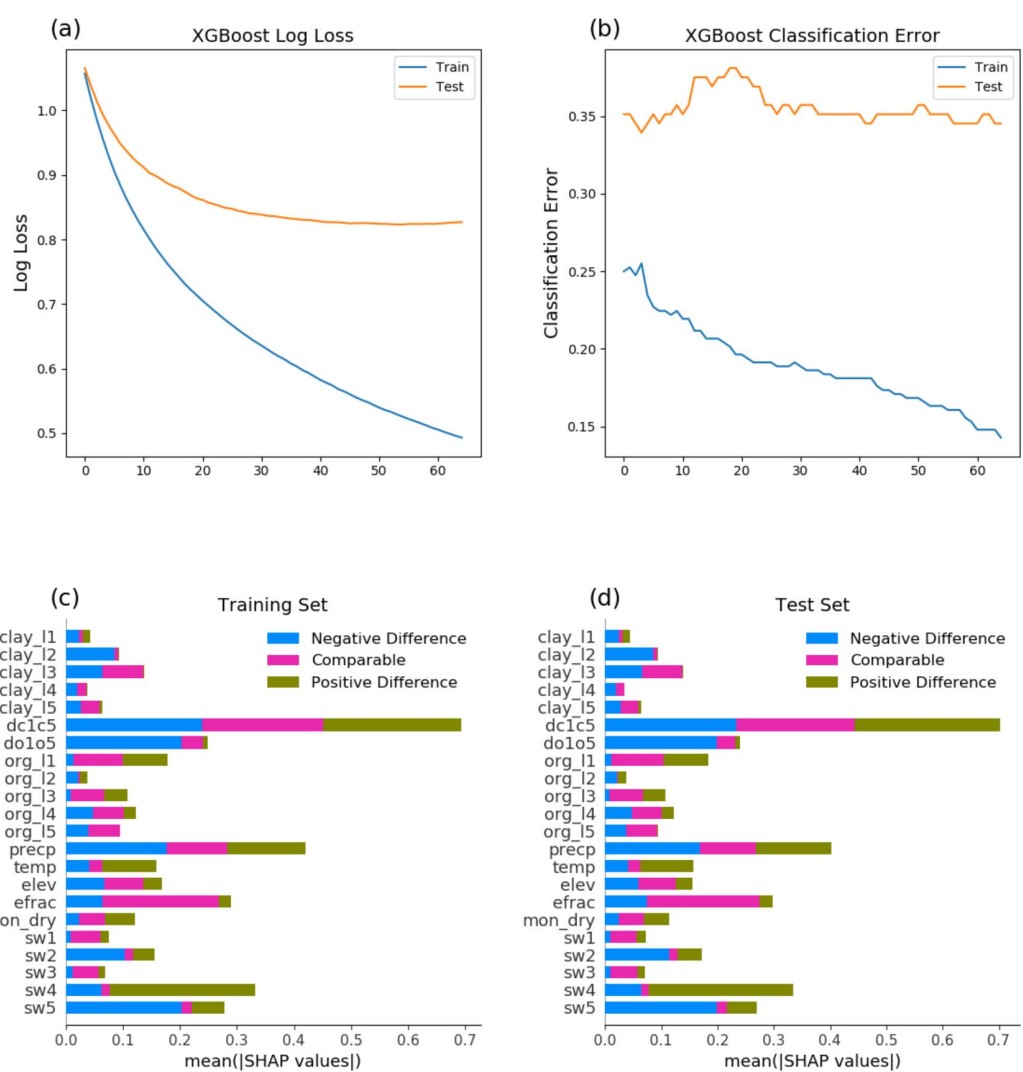

**Figure 5**. XGBoost model evaluation using selected conditions as predictors: learning curve, logarithmic loss (a), learning curve, classification error (b), feature importance for the training set (c), and feature importance for the test set (d)



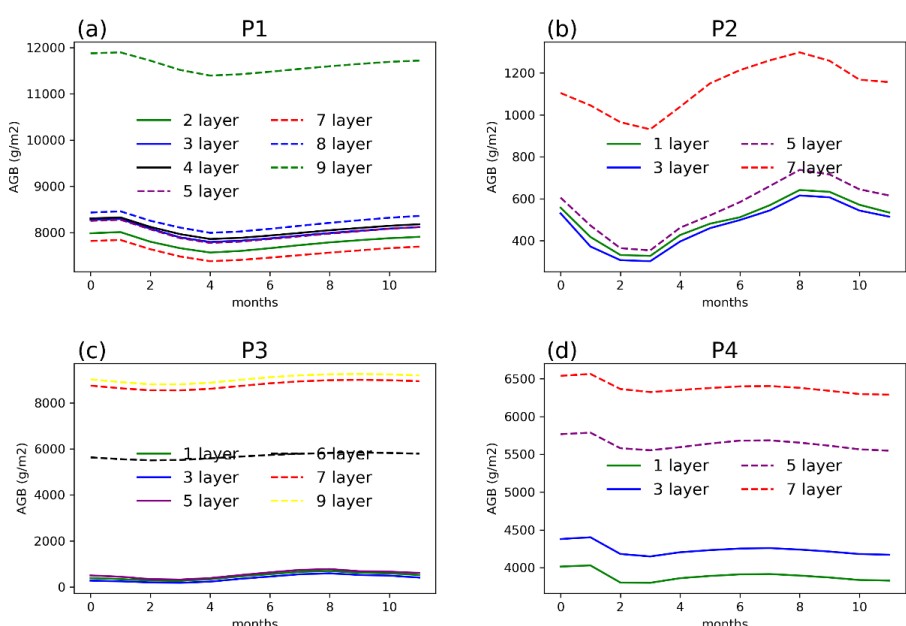

**Figure 6**. AGB from single point simulations at selected locations (P1 – P4) at year 100 of the simulations.

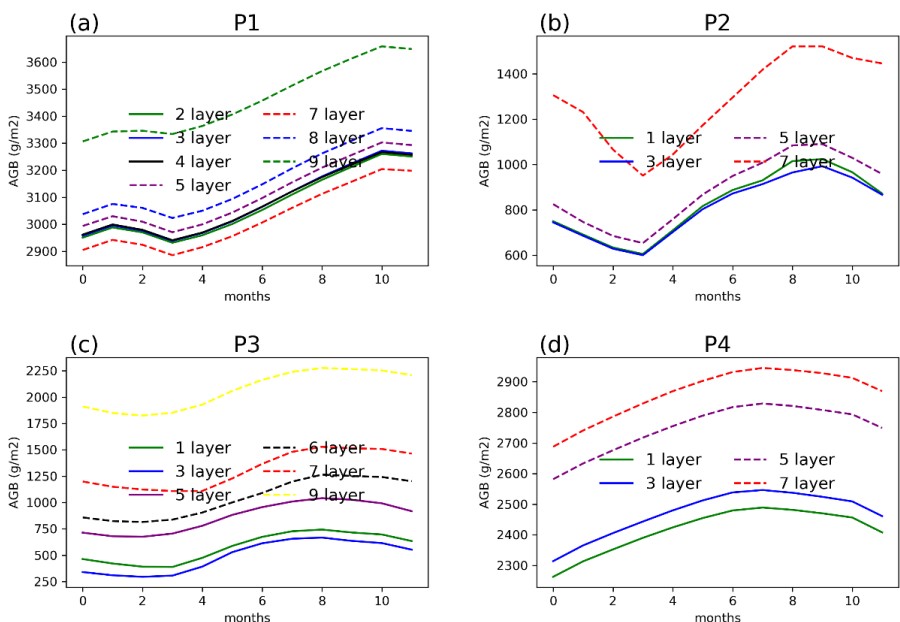


**Figure 7**. AGB from single point simulations at each selected location (P1 – P4) at year 10 of the simulations.



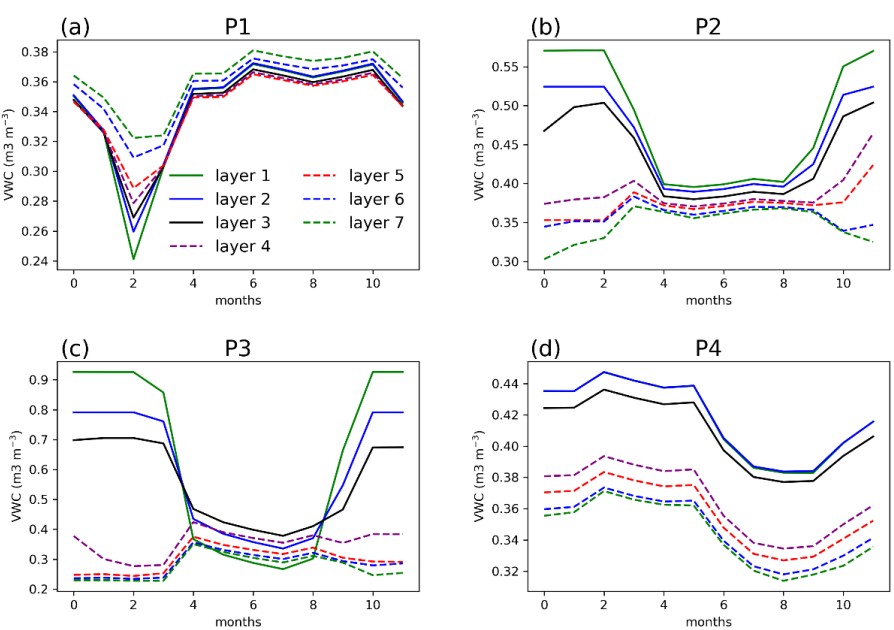

**Figure 8**. Volumetric water content (VWC) at selected points for single point simulations at 100 year of the simulation with no layer aggregation

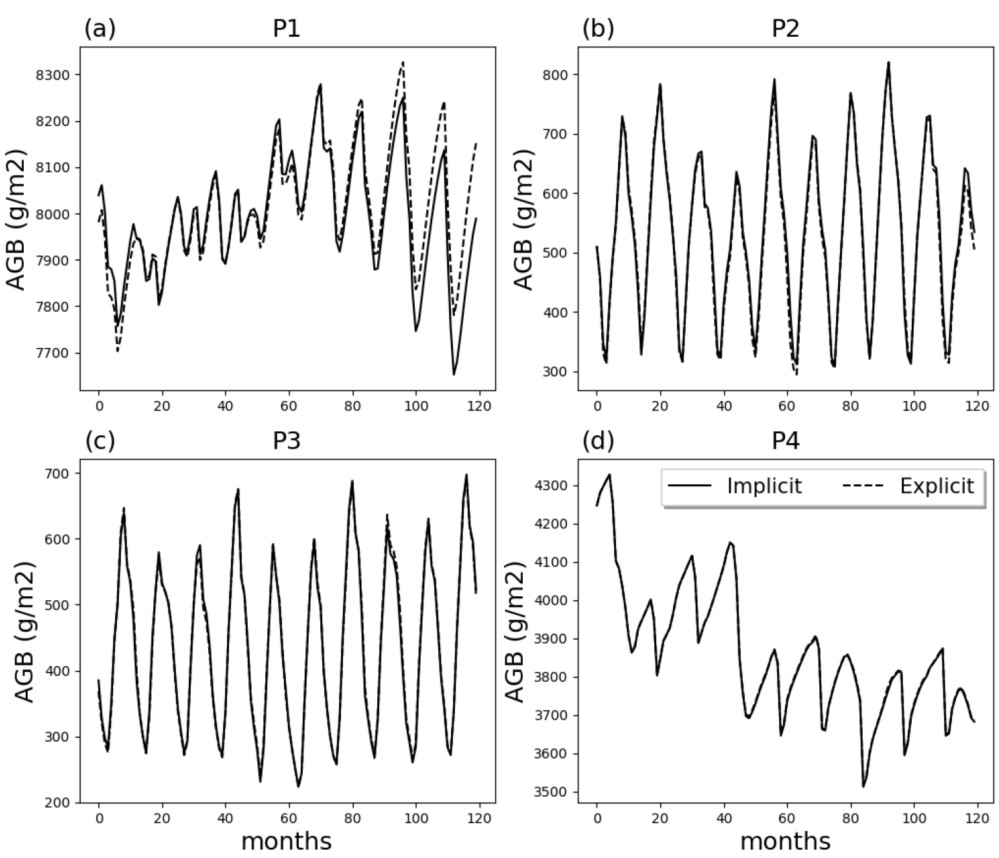

**Figure 9**. Comparison of AGB in the last 10 simulation years at points P1 to P4 with implicit and explicit integration methods.
