# Peer review of "Impact of numerical solution approach of a plant hydrodynamic model (v0.1) on vegetation dynamics"

_Geoscientific Model Development, 2022_

## Author Comment (AC1)

**We thank Professor Bohrer for the positive comments and suggestions.**

The development of FATES-HYDRO is important and represents an advance in modeling capability.

The study is conducted well, the code is made available through Zenoto, and the analysis is clear.

I have few minor comments that would help improve the comprehension of the results

Please add explicit vertically resolved formulation of how the soil interacts with the root. As is, the description is rather confusing (I could not figure out lines 220-225, or what "The stack of vertical soil-root interaction layers" at L190 means). I do not expect all the formulation of FATES to be repeated here, but the soil-root water interaction is the key physical process studied here, so at least that component of the formulation should be detailed to completion.

We apologize for the confusion. The following figure with explicit compartment numbers is used to illustrate how soil interacts with the roots. In this figure, the roots interact with a total of 10 soil layers. Compartment 1 represents leaf, 2 is stem, 3 is transporting root, 4, 10, ..., 58 are absorbing roots in soil layer 1, 2, ..., and 10, respectively. Each soil shell layer is divided into 5 compartments, with the innermost compartment (i.e., 5,11,...,59) directly interfacing with the absorbing root in each layer.

The discretized mass balance equation for each compartment becomes:

 $\rho_w V_1 \frac{d\theta_1}{dt} = Q_{1,2} - E$  , for compartment 1, Q1,2 is positive when flux is towards the atmosphere

 $\rho_w V_2 \frac{d\theta_2}{dt} = Q_{2,3} - Q_{1,2}$  , for compartment 2

 $\rho_w V_3 \frac{d\theta_3}{dt} = Q_{3,4} + Q_{3,10} + Q_{3,16} + Q_{3,22} + Q_{3,28} + Q_{3,34} + Q_{3,40} + Q_{3,46} + Q_{3,52} + Q_{3,58} - Q_{2,3} \text{ , for compartment 3}$

 $ho_w V_4 rac{d heta_4}{dt} = Q_{4,5} - Q_{3,4}$  , for compartment 4

 $\rho_w V_5 \frac{d\theta_5}{dt} = Q_{5,6} - Q_{4,5}$ , for compartment 5 and similarly for compartments 6,7, and 8

 $ho_w V_9 rac{d heta_9}{dt} = -Q_{8,9}$  , for compartment 9

Equation formulations for compartments 10 to 63 in the rest soil layers are the same as those corresponding compartments of 4 to 9 in the top layer.

When aggregated, total number of compartments is reduced by (number of layers aggregated -1) x 6. For example, when the top two layers are aggregated, compartments 58

to 63 disappear, and the sizes of the new compartments 4 to 9 are the combination of the old compartments 4 and 10, 5 and 11, and so on.

Figure R1. Example discretization of FATES-hydro

Also, list how betta (water stress factor) enters the transpiration/stomatal conductance calculation.

The stress factor modifies the top of canopy leaf photosynthetic capacity and the Ball-Berry leaf stomatal conductance as shown in Eqs. R1 and R2 below:

$$V_{c,max} = \beta V_{c,max} \tag{R1}$$

$$g_s = m \frac{A_n}{C_s / P_{atm}} h_s + \beta b \tag{R2}$$

where  $V_{c,max}$  is the maximum rate of carboxylation (µmol CO2 m-2 s-1),  $g_s$  is the leaf stomal conductance (µmol m-2 s-1), *m* is a plant functional type dependent parameter,  $A_n$  is leaf net photosynthesis (µmol CO2 m-2 s-1),  $C_s$  is the leaf surface CO2 partial pressure (Pa),  $P_{atm}$  is the atmospheric pressure (Pa),  $h_s$  is the leaf surface humidity, and *b* is the minimum stomatal conductance (µmol m-2 s-1),  $\beta$  is the stress factor defined by Eq. 4 in the manuscript.

You treated above ground biomass as the only tested indicator of model performance differences. I am very curious about other model related predictions, specifically, evapotranspiration and water use efficiency. Can you add some analysis of differences regarding these?

Thanks for the suggestion. The comparison of evapotranspiration (ET) and water use efficiency (WUE) are shown below for single point simulations. WUE is defined as the ratio of gross primary productivity (GPP) and ET. In general, the impact of grid aggregation on ET and WUE are small compared to that on AGB.

Figure R2. Evapotranspiration from single point simulations at selected locations (P1 - P4) at year 100 of the simulations.

---

## Author Comment (AC2)

**We thank the reviewer for the constructive comments and suggestions.**

The manuscript investigates how vertical resolution of soil-plant hydraulics and integration schemes influence a hydrodynamics-enabled biosphere model, FATES-HYDRO. The study conducted simulations by combining different numbers of top soil layers to create a gradient of different resolutions. They also use point-level simulations to explore the impacts of integration schemes.

Overall, I think the topic can be useful to the plant hydraulics and ecohydrological modeling community although I feel the design, interpretation, and presentation of the study can be further improved.

First, I think the underlying pathways of AGB changes under different resolutions are still elusive to me. Mixing top soil layers will surely influence hydraulic properties (as suggested by Fig. 4) but can also change the plant water accessibility right?

Mixing top soil layers will not change the total root biomass that can access water. However, it may change the solution of leaf water potential, thus the stress factor defined by Eq. 4 in the manuscript. It also triggers hydraulic failure mortality when a certain threshold is met. Hydraulic failure mortality begins when plant fractional loss of conductivity (ftc) reaches a threshold (ftc,t, default is 0.5):

$$M_{hf,coh} = \begin{cases} \frac{ftc-ftc,t}{1-ftc,t} m_{ft} & \text{for } ftc \ge ftc,t \\ 0.0 & \text{for } ftc < ftct \end{cases}$$
(R1)

where  $m_{ft}$  is the maximum mortality rate (yr-1).

The stress factor modifies the top of canopy leaf photosynthetic capacity and the Ball-Berry leaf stomatal conductance as shown in Eqs. R2 and R3 below:

$$V_{c,max} = \beta V_{c,max} \tag{R2}$$

$$g_s = m \frac{A_n}{C_s / P_{atm}} h_s + \beta b \tag{R3}$$

where  $V_{c,max}$  is the maximum rate of carboxylation (µmol CO2 m-2 s-1),  $g_s$  is the leaf stomal conductance (µmol m-2 s-1), m is a plant functional type dependent parameter,  $A_n$  is leaf net photosynthesis (µmol CO2 m-2 s-1),  $C_s$  is the leaf surface CO2 partial pressure (Pa),  $P_{atm}$  is the atmospheric pressure (Pa),  $h_s$  is the leaf surface humidity, and b is the minimum stomatal conductance (µmol m-2 s-1),  $\beta$  is the stress factor defined by Eq. 4 in the manuscript.

AGB changes due to the above modifications due to the change in numerical solution.

I am not sure how FATES calculate the soil-to-root conductivity but I guess root biomass/area matters? How big an effect this can be, especially if the distribution of root biomass is exponential?

Yes, root biomass/area matters for the soil-to-root conductivity. In FATES, soil-to-root conductivity is proportional to the root fraction in each layer, i.e., the longer the root in a layer, the larger the conductivity.

Furthermore, does soil moisture influence AGB mainly by influencing growth or mortality, which ultimately drives equilibrium biomass? Would be helpful to plot the difference of (relative) growth/mortality if they are in the standard output

Thanks for the suggestions. Soil moisture influence both growth and mortality as they are coupled.

Second, I am not sure how much I can trust the XGBoost analysis especially since the outof-sample accuracy is 67% (just a little different from random...). I guess including some variables on plants can help? (for example, average plant hydraulic traits within each grid cell?) In addition, using soil water potential rather than soil water might be better when looking at biomass differences...

**Thanks for the suggestions. We'll consider how to improve the model.**

Third, the AGB responses to the number of soil layers seem to be nonlinear and not necessarily monotonic in most of the 4 point-simulation sites (Fig.6). Why would this happen? Maybe some analysis of this point-level simulations can shed light upon large scale patterns.

Thanks for the comment. We don't expect the response of AGB to number of soil layers to be linear because of the nonlinearity of soil water retention curve and plant vulnerability curve and different layer soil properties.

Finally, I find the integration scheme analysis is simplistic and weak. For example, does a longer time step with explicit integration is computationally more efficient with a reasonable loss of accuracy? What would be the longest tolerable time step for plant hydraulics? How about other integration schemes such as Runge-Kutta? Such tests do not need to be long, I guess a few weeks worth of simulation is good enough so global simulations with different integration schemes might be possible.

**Thanks for the suggestion. We will consider further analysis.**

A few minor comments:

Line 165-200, this section is not easy to read with many parameters and poorly formatted equations, and some typos (e.g. in eq. 8, the higher order term should be o(delta^2) instead of 0). Please consider having a full editorial check and improve the readability.

**Thanks for catching the typos. We'll correct and improve the readability of the equations.**

Line 250, negative delta\_AGB --> overestimate reads very unintuitive. Please use experiment - reference simulations when calculating delta values.

Line 255, what is soil water saturation? Is it relative soil water?

Soil water saturation is the volume of water divided by volume of voids in the soil.

Figures:

Fig1 and Fig2 can be combined together since they both talks about vertical soil columns

Fig. 5, what are X axes in panels (a) and (b)? # of trees?

X axes in panels (a) and (b) are number of iterations or epochs for training.

---

## Author Response (AR1)

Dear Editor,

We want to express our appreciation of your time and effort handling the peer review of our manuscript.

We thank both reviewers for their positive and constructive comments that help improve the clarity of our manuscript. Attached please find our detailed point-by-point response to all reviewers' comments and the marked-up manuscript for the changes we made.

We look forward to hearing from you.

Sincerely,

Yilin Fang and Co-authors

**Response to RC1**

We thank Professor Bohrer for the positive comments and suggestions.

The development of FATES-HYDRO is important and represents an advance in modeling capability.

The study is conducted well, the code is made available through Zenoto, and the analysis is clear.

I have few minor comments that would help improve the comprehension of the results

Please add explicit vertically resolved formulation of how the soil interacts with the root. As is, the description is rather confusing (I could not figure out lines 220-225, or what "The stack of vertical soil-root interaction layers" at L190 means). I do not expect all the formulation of FATES to be repeated here, but the soil-root water interaction is the key physical process studied here, so at least that component of the formulation should be detailed to completion.

We apologize for the confusion. The following figure with explicit compartment numbers is used to illustrate how soil interacts with the roots and added in the Supplement. In this figure, the roots interact with a total of 10 soil layers. Compartment 1 represents leaf, 2 is stem, 3 is transporting root, 4, 10, …, 58 are absorbing roots in soil layer 1, 2, …, and 10, respectively. Each soil shell layer is divided into 5 compartments, with the innermost compartment (i.e., 5,11,…,59) directly interfacing with the absorbing root in each layer.

The discretized mass balance equation for each compartment becomes:

$\rho_w V_1 \frac{d\theta_1}{dt} = Q_{1,2} - E$ , for compartment 1, $Q_{1,2}$ is positive when flux is towards the atmosphere

$\rho_w V_2 \frac{d\theta_2}{dt} = Q_{2,3} - Q_{1,2}$ , for compartment 2

$\rho_w V_3 \frac{d\theta_3}{dt} = Q_{3,4} + Q_{3,10} + Q_{3,16} + Q_{3,22} + Q_{3,28} + Q_{3,34} + Q_{3,40} + Q_{3,46} + Q_{3,52} + Q_{3,58} - Q_{2,3}$ , for compartment 3

$\rho_w V_4 \frac{d\theta_4}{dt} = Q_{4,5} - Q_{3,4}$ , for compartment 4

$\rho_w V_5 \frac{d\theta_5}{dt} = Q_{5,6} - Q_{4,5}$ , for compartment 5 and similarly for compartments 6,7, and 8

$\rho_w V_9 \frac{d\theta_9}{dt} = -Q_{8,9}$ , for compartment 9

Equation formulations for compartments 10 to 63 in the rest of the soil layers are the same as those corresponding compartments of 4 to 9 in the top layer.

When aggregated, the total number of compartments is reduced by (number of layers aggregated -1) x 6. For example, when the top two layers are aggregated, compartments 58 to 63 disappear, and the sizes of the new compartments 4 to 9 are the combination of the old compartments 4 and 10, 5 and 11, and so on.

[Figure]

**Figure R1**. Example discretization of FATES-hydro

Also, list how betta (water stress factor) enters the transpiration/stomatal conductance calculation.

The stress factor modifies the top of canopy leaf photosynthetic capacity and the Ball-Berry leaf stomatal conductance as shown in Eqs. R1 and R2 below:

$$V_{c,max} = \beta V_{c,max} \tag{R1}$$

$$g_s = m \frac{A_n}{C_s/P_{atm}} h_s + \beta b \tag{R2}$$

where $V_{c,max}$ is the maximum rate of carboxylation (μmol $CO_2$ m$^{-2}$ s$^{-1}$), $g_s$ is the leaf stomatal conductance (μmol m$^{-2}$ s$^{-1}$), $m$ is a plant functional type dependent parameter, $A_n$ is leaf net photosynthesis (μmol $CO_2$ m$^{-2}$ s$^{-1}$), $C_s$ is the leaf surface $CO_2$ partial pressure (Pa), $P_{atm}$ is the atmospheric pressure (Pa), $h_s$ is the leaf surface humidity, and $b$ is the minimum

stomatal conductance (μmol m$^{-2}$ s$^{-1}$), $β$ is the stress factor defined by Eq. 4 in the manuscript. We added the above description in the revision.

You treated above ground biomass as the only tested indicator of model performance differences. I am very curious about other model related predictions, specifically, evapotranspiration and water use efficiency. Can you add some analysis of differences regarding these?

Thanks for the suggestion. We compared the model predictions of ET and water use efficiency (WUE) for the global simulation (Fig. R2) and single point simulations (Figs R3 and R4) due to layer aggregations. WUE is defined as the ratio of gross primary productivity (GPP) and ET. Compared to AGB, Layer aggregation has more impact on ET in the northern hemisphere (Fig. R2e), but the impact on WUE (Fig. R2f) is overall small globally.

From the single point simulations, the impacts of grid aggregation on ET and WUE (Figs. R3 and R4) are small compared to that on AGB in general, and the largest impact is at site P3. These comparisons are included in the revision and Supplement.

[Figure]

**Figure R2**. Model difference resulted from layer aggregations: percent change of AGB (Experiment – Reference) (a) and percent change of ET (b), and average soil water saturation between soil layer 1 and layer 5 in simulation year 100 (c) for the reference simulation, relative change of growth compared to the relative change of mortality (d), relative change of ET compare to the relative change of AGB (e), and relative change of WUE compared to the relative change of AGB (f). The pixels in white on land have values beyond the limits of the legends, associated with AGB < 0.5 gC m-2. Pixels with symbol × have ΔAGB less than 5%.

[Figure]

**Figure R3**. Evapotranspiration from single point simulations at selected locations (P1 – P4) at year 100 of the simulations.

[Figure]

**Figure R4**. Annual water use efficiency (WUE) from single point simulations at selected locations (P1 – P4) during the last 10 years of the simulations.

**Response to RC2**

We thank the reviewer for the constructive comments and suggestions.

The manuscript investigates how vertical resolution of soil-plant hydraulics and integration schemes influence a hydrodynamics-enabled biosphere model, FATES-HYDRO. The study conducted simulations by combining different numbers of top soil layers to create a gradient of different resolutions. They also use point-level simulations to explore the impacts of integration schemes.

Overall, I think the topic can be useful to the plant hydraulics and ecohydrological modeling community although I feel the design, interpretation, and presentation of the study can be further improved.

Thanks for the positive comments.

First, I think the underlying pathways of AGB changes under different resolutions are still elusive to me. Mixing top soil layers will surely influence hydraulic properties (as suggested by Fig. 4) but can also change the plant water accessibility right?

Mixing top soil layers will not change the total root biomass that can access water. However, it may change the solution of leaf water potential, thus the stress factor defined by Eq. 4 in the manuscript. It also triggers hydraulic failure mortality when a certain threshold of loss of conductivity is met. Hydraulic failure mortality begins when plant fractional loss of conductivity ($f_{tc}$) reaches a threshold ($f_{tc,t}$, default is 0.5):

$$M_{hf,coh} = \begin{cases} \frac{f_{tc}-f_{tc,t}}{1-f_{tc,t}} m_{ft} & \text{for } f_{tc} \geq f_{tc,t} \\ 0.0 & \text{for } f_{tc} < f_{tc,t} \end{cases} \qquad \text{(R1)}$$

where $m_{ft}$ is the maximum mortality rate (yr$^{-1}$), $f_{tc}$ is the maximum of $(1 - k_{r,i})$ for $i$ in plant compartments, $k_{r,i}$ is defined in Eq. 3 in the main text.

The stress factor modifies the top of canopy leaf photosynthetic capacity and the Ball-Berry leaf stomatal conductance as shown in Eqs. R2 and R3 below:

$$V_{c,max} = \beta V_{c,max} \qquad \text{(R2)}$$

$$g_s = m \frac{A_n}{C_s/P_{atm}} h_s + \beta b \qquad \text{(R3)}$$

where $V_{c,max}$ is the maximum rate of carboxylation (μmol $CO_2$ m$^{-2}$ s$^{-1}$), $g_s$ is the leaf stomatal conductance (μmol m$^{-2}$ s$^{-1}$), $m$ is a plant functional type dependent parameter, $A_n$ is leaf net photosynthesis (μmol $CO_2$ m$^{-2}$ s$^{-1}$), $C_s$ is the leaf surface $CO_2$ partial pressure (Pa), $P_{atm}$ is the atmospheric pressure (Pa), $h_s$ is the leaf surface humidity, and $b$ is the minimum stomatal conductance (μmol m$^{-2}$ s$^{-1}$), $\beta$ is the stress factor defined by Eq. 4 in the

manuscript. AGB changes due to the above modifications caused by different numerical solutions. We added these descriptions in the revision.

I am not sure how FATES calculate the soil-to-root conductivity but I guess root biomass/area matters? How big an effect this can be, especially if the distribution of root biomass is exponential?

Yes, root biomass/area matters for the soil-to-root conductivity. In FATES, soil-to-root conductivity is proportional to the root fraction in each layer, i.e., the longer the root in a layer, the larger the conductivity.

Furthermore, does soil moisture influence AGB mainly by influencing growth or mortality, which ultimately drives equilibrium biomass? Would be helpful to plot the difference of (relative) growth/mortality if they are in the standard output

Thanks for the suggestions. Using diameter growth increment (DDBH) to represent growth, we plotted the difference between the absolute percentage increase of growth and absolute percentage increase of mortality and found mixed influence of growth and mortality on AGB due to soil moisture (Fig. R1d). There are no specific patterns, but the influence on growth is greater than mortality in most pixels.

[Figure]

**Figure R1**. Model difference resulted from layer aggregations: percent change of AGB (Experiment – Reference) (a) and percent change of ET (b), and average soil water saturation between soil layer 1 and layer 5 in simulation year 100 (c) for the reference simulation, relative change of growth compared to the relative change of mortality (d), relative change of ET compare to the relative change of AGB (e), and relative change of WUE compared to the relative change of AGB (f).  The pixels in white on land have values beyond the limits of the legends, associated with AGB < 0.5 gC m-2. Pixels with symbol × have ΔAGB less than 5%.

Second, I am not sure how much I can trust the XGBoost analysis especially since the out-of-sample accuracy is 67% (just a little different from random...). I guess including some variables on plants can help? (for example, average plant hydraulic traits within each grid cell?) In addition, using soil water potential rather than soil water might be better when looking at biomass differences...

Thanks for the suggestions. We have 3 classes for this model. The theoretical baseline of random guessing for this problem is 38%. Our original model using soil water may not appear that satisfactory, but there is 30% improvement over the random guessing. The accuracy of the XGBoost model increased to 75% when we used soil water potential instead of soil water content to look at biomass differences. We didn't include plant hydraulic traits as they are not standard model output. We replaced the XGBoost model in the original submission with the one using soil water potential and made changes in description in the revision.

Third, the AGB responses to the number of soil layers seem to be nonlinear and not necessarily monotonic in most of the 4 point-simulation sites (Fig.6). Why would this happen? Maybe some analysis of this point-level simulations can shed light upon large scale patterns.

Thanks for the comment. We don't expect the response of AGB to the number of soil layers to be linear because of the nonlinearity of soil water retention curve and plant vulnerability curve and different layer soil properties, which will consequentially affect when growth or mortality will be more affected by changing soil water status. We added this statement in the revision.

Finally, I find the integration scheme analysis is simplistic and weak. For example, does a longer time step with explicit integration is computationally more efficient with a reasonable loss of accuracy? What would be the longest tolerable time step for plant hydraulics? How about other integration schemes such as Runge-Kutta? Such tests do not need to be long, I guess a few weeks worth of simulation is good enough so global simulations with different integration schemes might be possible.

Thanks for the suggestion. Because growth is a slow process, we did one-year experiments for the global simulation using 10-min and 30-min time steps and found even with the short 10-min time step, the AGB difference relative to the Reference case with implicit solve can be greater than 10% for some pixels as shown in Fig. R2. Note that FATES is part of an earth system model, which is expected to predict plant-soil hydraulic fluxes in innumerable conditions and extremes, over potentially long periods of time. The hydraulic solvers were therefore chosen based on the need to prioritize stability, which de-emphasizes the use of explicit solvers. We acknowledge there are other solvers that have been used effectively in hydraulic simulations (e.g., Crank-Nicholson, etc). There is often no best solver, but a decision on which solver to use has to be made. Having different solvers described in this study is fairly advanced compared to modeling of hydraulics in other earth system models and land models (ED2, etc). Testing more solver options would be nice, but would require incrementally more engineering and would exceed the scope of this manuscript. We added the above note in our revision.

[Figure]

**Figure R2**. Percent increase of AGB using 10-min time step explicit solve

A few minor comments:

Line 165-200, this section is not easy to read with many parameters and poorly formatted equations, and some typos (e.g. in eq. 8, the higher order term should be o(delta^2) instead of 0). Please consider having a full editorial check and improve the readability.

Thanks for catching the typo. We corrected the typo and carefully reviewed the equations to make sure they are correct. We also reformatted the equations to improve readability.

Line 250, negative delta_AGB --> overestimate reads very unintuitive. Please use experiment - reference simulations when calculating delta values.

Thanks for the suggestion. We added case names like "Reference case" and "Experiment case" and calculated delta values using Experiment – Reference and made changes throughout the manuscript in the revision.

Line 255, what is soil water saturation? Is it relative soil water?

Soil water saturation is the volume of water divided by the volume of voids in the soil.

Figures:

Fig1 and Fig2 can be combined together since they both talks about vertical soil columns

Thanks for the suggestion. We leave them as separate figures for the ease of descriptions.

Fig. 5, what are X axes in panels (a) and (b)? # of trees?

X axes in panels (a) and (b) are number of iterations or epochs for training. We added the label in the revision.

[revised manuscript text omitted]

$$\cancel{Q_{i,j} = -K_i\left(\rho_w g\left(z_i - z_j\right) + \left(\psi_i - \psi_j\right)\right)} \qquad (2)$$

$$Q_{i,j} = -K_i\left(\rho_w g\left(z_i - z_j\right) + \left(\psi_i - \psi_j\right)\right) \qquad (2)$$

The flux over a connection is driven by potential differences between compartments, where $g$ is acceleration due to gravity (9.81 m s⁻²) and $\psi_i$ is xylem or soil matric water potential (MPa), which is calculated based on pressure-volume curve, analogous to the soil water retention curve in ELM soil hydrology [*Christoffersen et al.*, 2016]; $z_i$ is the elevation above (positive) or below (negative) the ground (m), and $K_i$ is the conductance (kg Mpa⁻¹ s⁻¹) at the boundary between compartments $i$ and $j$. $K_i$ is calculated as the product of the relative hydraulic conductance $k_{r,i}$ (dimensionless) and the maximum conductance (kg mPa⁻¹ s⁻¹) at the boundary of nodes $i$. Note the maximum conductance is a product of the conduit cross-section and the material conductivity. Relative conductance or fraction of maximum conductance, $k_{r,i}$, is calculated by the vulnerability curve using an inverse polynomial function [*Manzoni et al.*, 2013] in plant compartment as follows:

$$\cancel{k_{r,i} = \left[1 + \left(\frac{\psi_i}{P_{50,i}}\right)^{a_i}\right]^{-1}} \qquad (3)$$

$$k_{r,i} = \left[1 + \left(\frac{\psi_i}{P_{50,i}}\right)^{a_i}\right]^{-1} \qquad (3)$$

$P_{50}$ is the water potential leading to 50% loss of hydraulic conductivity, $a_i$ is a shape index (dimensionless). The water stress function is usually empirically represented in land models as a function of soil water matric potential, but here is replaced by an empirical function of leaf water potential to include the hydraulic impacts on stomatal conductance [*Christofferson et al.* 2016]:

$$\beta = \left[1 + \left(\frac{\psi_t}{P_{50,gs}}\right)^{a_{gs}}\right]^{-1} \tag{4}$$

$$\beta = \left[1 + \left(\frac{\psi_l}{P_{50,gs}}\right)^{a_{gs}}\right]^{-1} \tag{4}$$

where $\beta$ is a water stress fraction, $\psi_l$ is the leaf water potential (MPa), $P_{50,gs}$ is the leaf water potential $\psi_l$ (MPa) at 50% stomatal closure, and $a_{gs}$ is the shape parameter (dimensionless). $\beta$ modifies the top of canopy leaf photosynthetic capacity and the Ball-Berry leaf stomatal conductance as shown in Eqs. 5 and 6 below:

$$V_{c,max} = \beta V_{c,max} \tag{5}$$

$$g_s = m\frac{A_n}{C_s/P_{atm}}h_s + \beta b \tag{6}$$

where $V_{c,max}$ is the maximum rate of carboxylation ($\mu$mol $CO_2$ m$^{-2}$ s$^{-1}$), $g_s$ is the leaf stomal conductance ($\mu$mol m$^{-2}$ s$^{-1}$), $m$ is a plant functional type dependent parameter, $A_n$ is leaf net photosynthesis ($\mu$mol $CO_2$ m$^{-2}$ s$^{-1}$), $C_s$ is the leaf surface $CO_2$ partial pressure (Pa), $P_{atm}$ is the atmospheric pressure (Pa), $h_s$ is the leaf surface humidity, and $b$ is the minimum stomatal conductance ($\mu$mol m$^{-2}$ s$^{-1}$), $\beta$ is the stress factor defined by Eq. 4.

Hydraulic failure induced mortality will be triggered when the plant fractional loss of conductivity ($f_{lc}$) reaches a threshold ($f_{lc,t}$, default is 0.5):

$$M_{hf,coh} = \begin{cases} \dfrac{f_{tc} - f_{tc,t}}{1 - f_{tc,t}} m_{ft} & \text{for } f_{tc} \geq f_{tc,t} \\ 0.0 & \text{for } f_{tc} < f_{tc,t} \end{cases} \qquad (7)$$

160   where $m_{ft}$ is the maximum mortality rate (yr$^{-1}$), $f_{tc}$ is the maximum of ($1 - k_{r,i}$) for $i$ in plant compartments, $k_{r,i}$ is defined in Eq. 3.

FATES-HYDRO divides each individual tree into four compartments: leaf, stem, transporting root (troot), and absorbing root (aroot) as shown in Figure 1.  In this study, all compartments except for the absorbing root are represented by a single node for each in the
165   discrete approximation of the equation. The absorbing root is discretized into the same number of nodes as the number of soil layers for soil hydrology in ELM.  The soil in each layer is radially discretized into cylindrical shells representing the rhizosphere around an absorbing root (Fig. 1). An example discretization with explicit compartment numbers is shown in Figure S1 in the Supplement and Eq. 1 for each compartment are listed in the Supplement as well to
170   demonstrate how each compartment interacts with the others, including the soil-root interaction.

[revised manuscript text omitted]

Note that the response of AGB to the number of soil layers aggregated is nonlinear because
of the nonlinearity of soil water retention curve and plant vulnerability curve and different layer
soil properties, which will consequentially affect when growth or mortality will be more affected
by the changing soil water status.

**3.3.2 Integration Methods**

Implicit and explicit integrations of Eq. 1 for points P1 to P4 were run to evaluate model
performance and computation costs. The simulations were performed without layer aggregation
for comparison of the integration schemes. The time step for the explicit integration is 10 min.
There are discrepancies between the two integration approaches at P1, but results show less than
2% AGB difference at the end of the simulation year (Fig. 9). Results at P2 to P4 are almost
identical. However, simulations took more time using the explicit integration approach, with wall
clock times 1.85, 1.31, 1.93, and 1.72 times of that of the implicit integration for P1 to P4,
respectively.

Note that FATES is part of an earth system model, which is expected to predict plant-soil
hydraulic fluxes in innumerable conditions and extremes, over potentially long periods of time.

The explicit approach is easier to implement than the implicit approach in terms of coding.

405 However, the explicit approach tends to have stability issue and requires small time steps, while the implicit approach is stable using large time steps but may require many iterations to converge to a solution. We acknowledge there are other solvers that have been used effectively in hydraulic simulations (e.g., Crank-Nicholson, etc.), but there is often no best solver. The hydraulic solvers in this study

410 were chosen based on the need to prioritize numerical stability for long simulations, which de-emphasizes the use of explicit solvers. The numerical experiments with different integration schemes in this study can serve as benchmark against each other. In the meantime, it shows that the 10-min time step in ED2 [*Xu et al.*, 2016] is a reasonable time step for these single point tests, but it is always a good practice to do convergence and stability tests for a specific study.

415 As a matter of fact, our one-year global simulation for the Reference case using the explicit integration and 10- min time step 
[revised manuscript text omitted]
 100 (c) for the Reference simulation, relative change of growth compared to the relative change of mortality (d), relative change of ET compared to the relative change of AGB (e), and relative change of WUE compared to the relative change of AGB (f). The pixels in white on land  have values beyond the limits of the legends, associated with AGB < 0.5 gC m-2. Pixels with symbol × have ΔAGB less than 5%.

[Figure]

[Figure]

**Figure 4**. Model differences resulted from layer aggregations: percent change of AGB (Experiment – Reference) (a), and E/ET (b) for simulation year 100, average clay content in the soil column (c), and dry season durations (months) (d), clay content difference (e) and organic matter difference (f) between layer 1 and the average of the top 5 layers from the surface. The pixels in white on land have values beyond the limits of the legends, associated with AGB < 0.5 gC m$^{-2}$. Pixels with symbol × have AGB differences less than 5%.

[Figure]

[Figure]

**Figure 5**. XGBoost model evaluation using selected conditions as predictors: learning curve, logarithmic loss (a), learning curve, classification error (b), feature importance for the training set (c), and feature importance for the test set (d)

[Figure]

**Figure 6**. AGB from single point simulations at selected locations (P1 – P4) at year 100 of the
625    simulations.

[Figure]

**Figure 7**. AGB from single point simulations at each selected location (P1 – P4) at year 10 of the simulations.

[Figure]

630

**Figure 8**. Volumetric water content (VWC) at selected points for single point simulations at 100 year of the simulation with no layer aggregation

[Figure]

**Figure 9**. Comparison of AGB in the last 10 simulation years at points P1 to P4 with implicit and explicit integration methods.